# Correlation between Anxiety Symptoms and Perception of Quality of Life in Women with More Than 24 Months after Undergoing Bariatric Surgery

**DOI:** 10.3390/ijerph19127052

**Published:** 2022-06-09

**Authors:** Jeane Lorena Dias Kikuchi, Manuela Maria de Lima Carvalhal, Ana Paula da Silva Costa, Jairisson Augusto Santa Brígida Vasconcelos, Carla Cristina Paiva Paracampo, Daniela Lopes Gomes

**Affiliations:** 1Postgraduate Program in Neuroscience and Behavior, Federal University of Pará, Belém 66075-110, PA, Brazil; cparacampo@gmail.com (C.C.P.P.); danielagomes@ufpa.br (D.L.G.); 2Faculty of Nutrition, Federal University of Pará, Belém 66075-110, PA, Brazil; manuela.carvalhall@gmail.com (M.M.d.L.C.); apsilvacosta97@gmail.com (A.P.d.S.C.); jairissonvasconcelos6@gmail.com (J.A.S.B.V.)

**Keywords:** obesity, bariatric surgery, quality of life and anxiety

## Abstract

Purpose: To analyze the correlation between anxiety symptoms and perceived quality of life in women more than 24 months after undergoing bariatric surgery. Methods: Cross-sectional, descriptive and analytical study, carried out with women who underwent bariatric surgery after at least 24 months. To assess the level of anxiety symptoms, the Beck Inventory was used and to assess the perception of quality of life, the Item Short Form Healthy Survey was applied. Results: Of the 50 participants, 36.0% had reports indicative of moderate symptoms and 64.0% had severe symptoms of anxiety. The domains of quality of life that correlated with better perception were pain (*p* < 0.001), functional capacity (*p* = 0.013), general health status (*p* = 0.018), social aspects (*p* < 0.001), and mental health (*p* < 0.001). In linear regression, a significant inverse correlation was found between the general emotional component of quality of life and anxiety score (β = −0.546; CI −1.419; −0.559; *p* < 0.001) and between the general physical component of quality of life and anxiety score (β = −0.339; CI −0.899; −0.131; *p* = 0.010), both independent of weight regain and surgery time. Conclusions: It was observed that moderate to severe anxiety symptoms seem to interfere with the perception of quality of life, regardless of weight regain and surgery time.

## 1. Introduction

Bariatric surgery today represents the most effective treatment for patients with severe obesity, making it possible to influence both weight loss and improved quality of life (QL) [1]. Currently, international guidelines recommend this procedure for individuals with a Body Mass Index (BMI) ≥ 40 kg/m^2^, as well as for those with a BMI ≥ 35 kg/m^2^ associated with co-morbidity [2].

According to the Brazilian Society of Bariatric and Metabolic Surgery in the year 2018, 70% of candidates for surgery were women, the most prevalent age group being between 35 and 50 years [3]. The procedure can lead to radical changes in the body and eating habits, causing emotional changes that can culminate in anxiety and depression, leading to repercussions on quality of life (QL), mainly affecting the female population [4].

QL is a condition that involves social, physical, and psychological aspects [5]. In the study by Akkayaoglu and Celik [6], the authors aimed to examine eating attitudes, perceptions of body image and QL of patients before and after bariatric surgery, in which an increase in QL was observed from the third month after surgery in both genders, mainly, in relation to the domains of physical and social functionality.

Le Foll et al. [7] noticed an increase in QL between 3 and 15 months post-bariatric and a significant decrease between 15 and 24 months after surgery. The study by Rolim et al. [8], when evaluating the QL of 42 people after 10 years of bariatric surgery, evidenced an improvement in 85.8% of the individuals.

An aspect to be considered as disadvantageous in the postoperative evolution is anxiety, as it can reflect on long-term weight gain after the surgical procedure, in addition to compromising the physical and mental QL of post-bariatric patients [9,10]. Della Méa and Peccin [11] in their study observed that 80% of the evaluated patients indicated the presence of minimal symptoms of anxiety and absence of severe symptoms between 12 and 24 months after the procedure.

Gill [12] performed a systematic review to assess changes in anxiety in postoperative bariatric surgery patients over 24 months. The authors found reductions in the overall severity of anxiety symptoms, and state that currently available evidence suggests that bariatric surgery is associated with long-term reductions in anxiety. However, there are still few investigations that have measured anxiety as a primary outcome. Therefore, it is not possible to state that bariatric surgery is an independent therapeutic tool for anxiety.

Baskaran [13] found depressive and anxiety symptoms in women before surgery, and when they were reassessed 6 months after the procedure, they noticed that these symptoms did not improve, despite significant weight loss, suggesting the need to investigate the determinants of the onset and the resolution of these symptoms in these patients. Patients in the postoperative period of bariatric surgery, when they have a prevalence of anxiety symptoms, are more dependent on nursing care, thus, anxiety negatively influences the organic evolution of patients [14].

In this context, several aspects related to physical health and mental health can be affected after bariatric surgery, highlighting issues related to anxiety symptoms that can be affected in the maintenance of weight loss and QL. Furthermore, knowing that most patients have improvements in their physical and mental conditions in the immediate postoperative period, it is relevant to assess whether in the long-term postoperative period, some portion of these patients still present beneficial or non-variable evolutions. Thus, considering that no studies were found that evaluated possible associations between anxiety and QL in women after 24 months of bariatric surgery, this article aimed to verify whether there is a correlation between perceived QL and anxiety symptoms, considering weight recurrence, in women more than 24 months after bariatric surgery.

## 2. Materials and Methods

### 2.1. Study Type

This is a cross-sectional, descriptive, and analytical study, carried out in a public hospital in Belém, Pará, with adult women who underwent bariatric surgery for at least 24 months. The study was approved by the research ethics committee of the Institute of Health Sciences of the Federal University of Pará (first opinion no. 2.170.863 and second opinion no. 3.329.834, for extension of data collection), complying with legal requirements pursuant to resolution 466 of 12 December 2012, of the National Health Council and according to the Helsinki Declaration.

The authors of this study do not present any conflicts of interest.

### 2.2. Participants

Non-probabilistic convenience sampling was performed. The study included women, aged between 18 and 59 years, who underwent bariatric surgery for at least 24 months, through surgical techniques of gastric bypass or sleeve and who agreed to participate in the research by signing the Informed Consent Form (ICF). All were invited to participate by telephone, through an extension project of a federal university in Belém, Pará, Brazil, which provides nutritional care for patients after bariatric surgery.

Women who underwent another type of surgery, who became pregnant after surgery, who were using medications that could affect the data analysis, who lived outside the metropolitan region of Belém/PA, and could not attend the research evaluation stages, were excluded.

### 2.3. Sex Inclusive Reporting

In this study, only women were included, as only three males were included in the sample. Thus, as body composition and energy expenditure are different between genders, and one of the variables studied was weight regain, men were excluded in order to avoid methodological bias.

### 2.4. Data Collect

Information on sociodemographic characteristics (age, income, education level, date of surgery, initial pre-surgical weight and lower weight achieved after surgery) was obtained. In the anthropometric assessment, body weight was measured using a Welmyr platform-type scale, with a capacity of 150 kg, with a coupled stadiometer (200 cm, precision of 1 mm), which was used to measure height, with the participants barefoot and wearing light clothes. From the measured weight and height, the BMI (BMI = Weight/Height^2^) was calculated, obtaining the classification of the nutritional status recommended by the World Health Organization [15]. Pre- and post-surgical weight values were also collected, considering the weight in the first preoperative visit, the weight in the week of surgery, and the lowest stable weight achieved after surgery.

The ideal weight and evaluation of the percentage of excess weight loss was calculated from the table of the Metropolitan Life Foundation [16], as suggested by the Brazilian Consensus on Bariatric Surgery [17]. The calculation of the percentage of weight regain was based on the difference between the current weight and the lowest stable weight achieved by the patient after surgery, with a weight regain of 15% considered as significant, as proposed by Lasagni et al. [18].

### 2.5. Instruments

The Medical Outcomes Study 36-Item Short-Form Health Survey (SF-36) was used to assess the perception of QL, a translated and validated version for Portuguese that uses 36 questions on various aspects related to perception and QL related to health. This instrument is subdivided into domains that correspond to: Physical Component (domains of functional capacity, limitation by physical aspects, pain, and vitality); and Emotional Component (mental health domains, general health status, limitation due to emotional aspects, and social aspects). The categories range from 2 to 10 items and all of them can be summarized in two components: Overall Score of Physical Components and Overall Score of Emotional Components. The results are expressed as a score, varying on a scale from 0 to 100, in which 0 corresponds to the worst perception and 100 to the best perception of QL [19].

The level of anxiety assessed through the Beck Inventory (BAI), which was translated and validated for Brazil by Cunha [20]. This is a self-report instrument consisting of 21 items with descriptive statements of anxiety symptoms. Items are calculated using a scale of four response options: 0—absolutely not; 1—mildly: it didn’t bother me too much; 2—moderately: it was very unpleasant, but I could bear it; and 3—severely: difficult to bear. The score was performed by the sum of individual scores ranging from 0 to 63. Anxiety symptoms were classified as: minimum level: scores from 0 to 10; mild level: scores from 11 to 19; moderate level: scores from 20 to 30; and severe level: scores from 31 to 63.

### 2.6. Data Analysis

For descriptive statistics, data were expressed through measures of central tendency and dispersion. Statistical tests were chosen according to the classification of variables and sample distribution. Spearman’s correlation test was used to test the correlation between the anxiety score and the perception of QL. Variables that showed a statistically significant correlation in the bivariate analysis were included in the linear regression model. IBM SPSS Statistics for Windows, Version 24.0 (IBM Corp., Armonk, NY, USA), considering the level of statistical significance of *p* < 0.05.

## 3. Results

In total, 50 women were evaluated. It is observed in Table 1 that the mean surgical time of the participants was 61.9 ± 47.2 months, 68% (*n* = 34) underwent gastric bypass and 32% (*n* = 16) underwent sleeve. As for anthropometry, there was an average preoperative BMI of 44.0 ± 6.6 kg/m^2^ and current BMI of 29.7 ± 5.4 kg/m^2^. As for the loss of excess weight, an average of 75.6 ± 28.8% was observed; 60% (*n* = 30) of the women had significant weight regain, with an average of 23.3 ± 18.4%.

Regarding the level of anxiety symptoms, 64% (*n* = 32) had severe symptoms and 36% (*n* = 18) moderate. Regarding the perception of QL, the domains with the best perception were in the aspect of functional capacity and limitation by physical aspects, demonstrating good quality perception in aspects related to displacement and physical activity. The worst perceptions were in the domains of pain and vitality, demonstrating pain during daily activities and low vigor and motivation.

Table 2 shows the correlation between the anxiety score and the domains of perceived quality of life. There was a correlation with the general score of physical components (Ρ^2^ = −0.407; *p* = 0.003), functional capacity (Ρ^2^ = −0.347; *p* = 0.013), pain (Ρ^2^ = −0.475; *p* < 0.001), and general health status (Ρ^2^ = −0.333; *p* = 0.018). As for the general score of emotional components (Ρ^2^ = −0.455; *p* = 0.001), it was observed that social aspects (Ρ^2^ = −0.637; <0.001) and mental health (Ρ^2^ = −0.524; *p* = <0.001) also correlated with the anxiety score.

The linear regression model, shown in Table 3, indicates that in model 1 there was a significant inverse correlation between the general emotional component of QL and anxiety score (β = −0.593; IC −1.498; −0.651; *p* < 0.001). In model 2, the inverse correlation between the general emotional component of QL and anxiety score remained independent of weight regain (β = −0.543; IC −1.408; −0.558; *p* < 0.001). In model 3, the inverse correlation between the general emotional component of QL and the anxiety score (β = −0.546; IC −1.419; −0.559; *p* < 0.001) was maintained, regardless of weight regain and surgery time.

Table 4 shows the linear regression model indicating that, in model 1, there was a significant inverse correlation between the general physical component of QL and the anxiety score (β = −0.417; IC −1.033; −0.232; *p* < 0.001). In model 2, the inverse correlation with the anxiety score remained independent of weight regain (β = −0.331; IC −0.886; −0.118; *p* < 0.001). In model 3, the inverse correlation between the general physical component of quality of life and anxiety score (β = −0.339; CI −0.899; −0.131; *p* = 0.010), both independent of weight regain.

## 4. Discussion

The present study evaluated the correlation between anxiety symptoms and perceived quality of life in women more than 24 months after undergoing bariatric surgery. Only women were evaluated because there is a higher prevalence of bariatric surgeries in this population, which can be justified by the fact that females are more concerned with appearance and aesthetic issues when compared to males [21]. Furthermore, in the study carried out by Ribeiro et al. [22] for example, it was found that 81% of the participants in the preoperative period of bariatric surgery were women and that they obtained higher scores in the classification of moderate anxiety compared to men, using the Beck Anxiety Inventory (BAI).

In this sense, the results showed that 64% and 36% of the participants had severe and moderate anxiety symptoms, respectively. Similar results were obtained by Ribeiro et al. [23] who observed that, between 24 and 59 months after surgery, 33% of patients had symptoms suggestive of anxiety, and 60 months after surgery, 40% of patients had some degree of anxiety. In addition, the authors pointed out that patients had improved anxiety levels in the first 23 months after surgery. Taken together, these results show that anxiety symptoms tend to be present in bariatric patients after 24 months of surgery. One hypothesis for this finding is that with the stabilization or recovery of body weight, anxiety symptoms may (re)appear, however, there is no way to establish a cause-and-effect relationship. That is, there is no way to assume whether patients present symptoms of anxiety caused by weight regain, or if anxiety symptoms may also contribute to weight gain.

In the present study, the domains with the best perception of QL were functional capacity and limitation due to physical aspects. These data complement the findings by Nickel et al. [24] who, when assessing the QL at 6 months and 24 months after surgery, concluded that bariatric surgery had a greater effect on QL due to physical aspects than on mental QL, which improved in the short term and began to decrease within 24 months after surgery. These data indicate that the increase in physical QL is related to the decrease in weight in the postoperative period, however, and as expected, if there is weight regain, the physical QL tends to decrease in these patients.

Further, considering that in our results the physical QL presented good scores, it is in agreement with the study carried out by Major et al. [25] on the effect of bariatric surgery on the long-term QL in 65 patients. The authors found that the perception of physical health QL increased significantly after the 10-year period. It is important to consider that although the literature has already established short-term improvement in QL after bariatric surgery, as in the results found by Sánchez-Piedrahita, Castañeda-Avilés and Núñez-Gómez [26] and Nickel et al. [24], there is still a need for further studies evaluating very long-term outcomes.

Additionally, the present study observed an inverse correlation between the perception of the general emotional component of QL and anxiety, regardless of weight regain and surgery time. This result is convergent with the one composed by Kalarchian et al. [27] in which, regardless of weight change, mood and anxiety disorders are related to a smaller improvement in the QL that is related to mental health in the long-term postoperative period. Together, these results obtained are necessary when aimed at improving QL, in order to face anxiety and other aspects of the emotional component in the context of long-term postoperative bariatric surgery. In the same context, Al Khalifa and Al Ansari [26,28] observed that the postoperative QL assessments of patients undergoing the sleeve surgical technique showed significant improvements in all items, except in the mental health domain.

A correlation was also observed between anxiety and low general scores of physical components and functional capacity, indicating that moderate and severe degrees of anxiety can interfere with the ability and autonomy to be more independent in daily activities, as this dimension intends to measure how much the state of health interferes with simple activities such as taking a shower, walking, climbing stairs, running, sweeping the floor, or carrying weight.

Unlike the data found in this study, Sockalingam et al. [29] when evaluating psychosocial predictors, such as anxiety and QL 2 years after bariatric surgery, found in their results that anxiety had no significant influence on health-related QL or weight. In addition, the study patients had a low rate of anxiety symptoms (15.1%). In this sense, Le Foll [7] highlighted the importance of monitoring around 15 to 18 months after bariatric surgery, as this period can be identified as the first “critical” period where weight regain and a decreased self-perception of QL may occur. A possible explanation for the divergence between the results of this research and those found by Sockalingam et al. [29] may be the different methods adopted in the two studies to assess symptoms of anxiety, and the fact that the sample in the study by Sockalingam et al. [29] included participants of both sexes.

It was also found that the worse the perception of the general physical component of QL, the greater the anxiety, regardless of weight regain and surgery time. This result differs from the one found by Freire et al. [30], who observed an association between anxiety symptoms and weight regain in about 86% of patients, between 7 to 14 years after surgery. Considering these findings, it can be assumed that the increase in anxiety symptoms over the years may interfere with the individual’s level of independence, affecting their basic activities and leaving them functionally dependent, which, consequently, will directly affect their QL.

As for anthropometry, a mean preoperative BMI of 44.0 ± 6.6 kg/m^2^ and current BMI of 29.7 ± 5.4 kg/m^2^ was observed in this study. Similar results were found by Reichmann et al. [31] who recorded very similar means of preoperative BMI (45.0 ± 6.54 kg/m^2^) and current BMI (31 ± 6.84 kg/m^2^). Regarding the surgical technique, most participants in this study underwent gastric bypass (68%). Bardal, Ceccatto, and Mezzomo [32] also found a prevalence of gastric bypass in their investigation, corroborating the data from the International Federation for the Surgery of Obesity and Metabolic Diseases (IFSO), which claims that the gastric bypass is the most performed surgical technique in Brazil, with about 76.6% [33].

The present study has some limitations that should be taken into account, such as the small sample size and the lack of follow-up of the women studied. However, despite this, it is worth noting that this study was carried out only in women, which helps to elucidate which variables have interfered with the QL of this profile of patients after 24 months of bariatric surgery, since most studies include both women and men, thus allowing us to identify which aspects should be taken into account in the follow-up in the long-term to prevent or treat symptoms of anxiety and occurrence of weight regain in order to preserve the improvement in QL achieved postoperatively for clinical practice. Furthermore, the variables used in this study, such as QL, anxiety, and weight regain, used only in women, were not listed in other investigations, which points to the need for further research in this area and the non-generalizability of these results.

## 5. Conclusions

It is concluded that anxiety symptoms seem to be common in women after 24 months of bariatric surgery, and may directly interfere with emotional and physical QL, even when controlling for weight regain and surgery time in the multivariate analysis, showing a strong relationship between these factors. Thus, the importance of long-term intervention by multidisciplinary teams is highlighted, as well as support networks, with a focus on anxiety symptoms in order to increase the QL of these individuals, which can compromise the success of the treatment, in particular after 24 months of surgery.

## Figures and Tables

**Table 1 ijerph-19-07052-t001:** Characterization of the clinical and anthropometric profile, anxiety symptoms, and perception of quality of life of women more than 24 months after undergoing bariatric surgery.

	Mean ± SD */*n* *	Interval/%
Surgery time (months)	61.9 ± 47.2	24–204
Surgical technique		
Gastric bypass	34	68.0
Vertical gastrectomy	16	32.0
Anthropometry		
Pre-operative Body Mass Index (kg/m^2^)	44.0 ± 6.6	32.9–57.5
Current Body Mass Index (kg/m^2^)	29.7 ± 5.4	20.8–43.7
Excess weight loss (%)	75.6 ± 28.8	0–137.5
Weight regain (%)	23.3 ± 18.4	0–81.0
Present	30	60.0
Absent	20	40.0
Anxiety symptoms level		
Minimum	0	0.0
Mild	0	0.0
Moderate	18	36.0
Severe	32	64.0
Quality of life perception		
Functional capacity	78.8 ± 18.1	25.0–100.0
Limitation by physical aspects	72.5 ± 37.2	0.0–100.0
Pain	56.6 ± 24.5	0.0–100.0
General health status	60.3 ± 15.8	25.0–87.0
Vitality	57.3 ± 20.7	0.0–90.0
Social aspects	68.5 ± 25.9	12.5–100.0
Limitation by emotional aspects	67.3 ± 42.9	0.0–100.0
Mental health	68.0 ± 17.0	28.0–100.0
General physical component	67.1 ± 16.7	21.3–92.8
General emotional component	65.3 ± 20.0	13.6–94.5

* SD = Standard deviation/*n* = number.

**Table 2 ijerph-19-07052-t002:** Correlation between the anxiety symptoms score and the perception of quality of life of women more than 24 months after undergoing bariatric surgery.

Anxiety Score	Ρ^2^	*p*-Value *
General physical components score	−0.407	0.003
Functional capacity	−0.347	0.013
Limitation by physical aspects	−0.013	0.930
Pain	−0.475	<0.001
General health status	−0.333	0.018
General emotional components score	−0.455	0.001
Vitality	−0.268	0.060
Social aspects	−0.637	<0.001
Limitation by emotional aspects	−0.212	0.139
Mental health	−0.524	<0.001

* Spearman correlation test.

**Table 3 ijerph-19-07052-t003:** Linear regression between the anxiety symptoms score and the general emotional component of the quality of life of women more than 24 months after undergoing bariatric surgery.

	*B*	CI 95% (Minimum; Maximum)	*p*-Value
Model 1			
Anxiety score	−0.593	−1.498; −0.651	<0.001
Model 2			
Anxiety score	−0.543	−1.408; −0.558	<0.001
Weight regain	−0.214	−0.881; 0.040	0.073
Model 3			
Anxiety score	−0.546	−1.419; −0.559	<0.001
Weight regain	−0.227	−0.926; −0.559	0.068
Surgery time	0.050	−0.083; 0.126	0.675

Notes: Linear regression; Dependent variable: General emotional component of quality of life; co-variable: General anxiety score, weight regain (kg) and surgery time (months), *B* = Coefficient of regression.

**Table 4 ijerph-19-07052-t004:** Linear regression between the anxiety symptoms score and the general physical component of the quality of life of women more than 24 months after undergoing bariatric surgery.

	*B*	CI 95% (Minimum; Maximum)	*p*-Value
Model 1			
Anxiety score	−0.417	−1.033; −0.232	0.003
Model 2			
Anxiety score	−0.331	−0.886; −0.118	0.011
Weight regain	−0.365	−1.016; −0.185	0.006
Model 3			
Anxiety score	−0.339	−0.899; −0.131	0.010
Weight regain	−0.400	−1.086; −0.228	0.003
Surgery time	0.135	−0.044; 0.143	0.294

Notes: Linear regression; Dependent variable: General physical component of quality of life; co-variable: General anxiety score, weight regain (kg) and surgery time (months), *B* = Coefficient of regression.

## Data Availability

The data presented in this study are available on request from the corresponding author.

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
