# Peer review of "Correlation between Anxiety Symptoms and Perception of Quality of Life in Women with More Than 24 Months after Undergoing Bariatric Surgery"

_ijerph, 2022, doi:10.3390/ijerph19127052_

Round 1

Reviewer 1 Report

  1. Abstract: P values should be shown as “p < 0.001” when it is smaller than 0.001. It should not be shown as “p=0.000”.

  1. Abstract: The statement “between the general physical component of quality of life and anxiety score (β = -0.339; CI -0.899; -0.131; p= 0.010), both independent of weight regain and surgery time.” appears to be misleading because surgery time was not significant (p = 0.294). The statement should be “between the general physical component of quality of life and anxiety score (β = -0.339; CI -0.899; -0.131; p= 0.010), both independent of weight regain.”

  1. Line 29 and 106’: “Body Mass Index” should not be capitalized.

  1. Line 42: “Lee Fool” should be “Le Foll”.

  1. Line 92: The word “confuse” should be replaced by “affect”.

  1. Line 142: “The Statistical Package for Social Science version 24.0 program” should be described as “IBM SPSS Statistics for Windows, Version 24.0 (Armonk, NY, IBM Corp.)”

  1. Table 1: In the header row, “DP” should be “SD”.

  1. Table 2 and line 162-166: The symbol for the Spearman’s correlation coefficient should be the Greek letter rho (ρ) and not p. In addition, it should not be squared.

  1. Line 172-177, 185-188, and Table 3: P values should be shown as “p < 0.001” when it is smaller than 0.001. It should not be shown as “p=0.000”.

  1. Table 3 and 4: The title should be “Linear regression” instead of “Correlation”.

  1. Data analysis: According to the footnote of Table 2, Spearman correlation test rather than Pearson’s correlation test was used. Please correct the inconsistence.

  1. Data analysis: Have the authors evaluated if “current BMI” was a significant factor in the linear regression models in Table 3 and 4?

  1. References: Please check all the references for accuracy and format. For example, reference #8 the page number should be e1916. Reference #26: the page number should be 44. In addition, reference #11: The names of the two authors should be Pilla Della Méa, C., & Peccin, C. See: https://pssaucdb.emnuvens.com.br/pssa/article/view/370

Author Response

International Journal of Environmental Research and Public Health- Special Issue "Effects of Stress Exposure on Mental Health and Well-Being".

It is with great pleasure that we submit for review the manuscript entitled “Correlation between anxiety symptoms and perception of quality of life in women with more than 24 months after undergoing bariatric surgeryfor consideration by International Journal of Environmental Research and Public Health

We appreciate the reviewer(s) suggestions and clarify that all requested changes have been made. According to the instructions:

Reviewer: 1

  • Abstract: P values should be shown as “p < 0.001” when it is smaller than 0.001. It should not be shown as “p=0.000”.

R= We appreciate the correction and inform you that the changes have been made.

  • Abstract: The statement “between the general physical component of quality of life and anxiety score (β = -0.339; CI -0.899; -0.131; p= 0.010), both independent of weight regain and surgery time.” appears to be misleading because surgery time was not significant (p = 0.294). The statement should be “between the general physical component of quality of life and anxiety score (β = -0.339; CI -0.899; -0.131; p= 0.010), both independent of weight regain.”

R= We thank you for your consideration. In the linear regression test (Table 4), the general physical component of quality of life was considered as the dependent variable and the anxiety score, weight regain and surgery time as independent variables.

Therefore, it was observed that the correlation between anxiety and the general component remained independent of weight regain (p=0.011) and surgery time (p=0.010), even though no correlation was observed between weight regain and general physical component of quality of life, therefore, we did not make this change.

  • Line 29 and 106’: “Body Mass Index” should not be capitalized.

 R= Thanks for the suggestion, changes have been made to the indicated lines.

  • Line 42: “Lee Fool” should be “Le Foll”.

R= Thank you for the correction, the requested adjustment has been made.

  • Line 92: The word “confuse” should be replaced by “affect”.

R= We appreciate the comment, the adjustment has been made.

  • Line 142: “The Statistical Package for Social Science version 24.0 program” should be described as “IBM SPSS Statistics for Windows, Version 24.0 (Armonk, NY, IBM Corp.)”

 R=  We appreciate the correction and inform you that the change has been made.

  • Table 1: In the header row, “DP” should be “SD”.

 R=  Thanks for the correction and we apologize for the error. Change was made.

  • Table 2 and line 162-166: The symbol for the Spearman’s correlation coefficient should be the Greek letter rho (ρ) and not p. In addition, it should not be squared.

 R=  We appreciate the corrections and inform you that the adjustments have been made.

  • Line 172-177, 185-188, and Table 3: P values should be shown as “p < 0.001” when it is smaller than 0.001. It should not be shown as “p=0.000”.

R=We appreciate the corrections and inform you that the changes have been made.

  • Table 3 and 4: The title should be “Linear regression” instead of “Correlation”.

R=Thanks for the suggestions, the changes have been made.

  1. Data analysis: According to the footnote of Table 2, Spearman correlation test rather than Pearson’s correlation test was used. Please correct the inconsistence.

R= Thank you for your consideration, correction has been made.

  • Data analysis: Have the authors evaluated if “current BMI” was a significant factor in the linear regression models in Table 3 and 4?

R= We appreciate the suggestion. The BMI was tested in the model, but there was no change in the correlation. In addition, we emphasize that BMI was not maintained in the linear regression, as this is a cross-sectional study, and BMI is data that is influenced by other variables, such as initial weight. Therefore, it would be more interesting to use BMI if it were a prospective study from the preoperative time. So, for the present cross-sectional study, with the bariatric surgery population, postoperative weight regain seems to have a greater influence on issues such as mental health and quality of life.

  • References: Please check all the references for accuracy and format. For example, reference #8 the page number should be e1916. Reference #26: the page number should be 44. In addition, reference #11: The names of the two authors should be Pilla Della Méa, C., & Peccin, C. See: https://pssaucdb.emnuvens.com.br/pssa/article/view/370

R= Thanks for the corrections, the changes have been made.

Reviewer 2 Report

The article analyses the correlation between anxiety symptoms and perceived quality of 10 life in women with more than 24 months after undergoing bariatric surgery. As other studies point out, participants had severe and moderate anxiety symptoms (respectively, in this case, 64% and 36%).

Furtherly, the study demonstrates that moderate to severe anxiety symptoms interfere with the perception of quality of life, regardless of weight regain and surgery time.

Evidence of this article are relevant in highlighting the need of an adequate care by multidisciplinary teams, as well as support networks, in order to prevent that anxiety symptoms, together with a decrease of the quality life improvements, can compromise the success of the treatment in the long-term, in particular after 24 months of surgery.

As the authors insist (7, 267) that the results of the research are related to the total female sample of the study participants, some more information on possible gender-specific factors that could influence the results of the research topics would be useful.

Finally, I think “Pezzim” - before [14] - is a typo (2, 64).

The article analyses the correlation between anxiety symptoms and perceived quality of 10 life in women with more than 24 months after undergoing bariatric surgery. As other studies point out, participants had severe and moderate anxiety symptoms (respectively, in this case, 64% and 36%).

Furtherly, the study demonstrates that moderate to severe anxiety symptoms interfere with the perception of quality of life, regardless of weight regain and surgery time.

Evidence of this article are relevant in highlighting the need of an adequate care by multidisciplinary teams, as well as support networks, in order to prevent that anxiety symptoms, together with a decrease of the quality life improvements, can compromise the success of the treatment in the long-term, in particular after 24 months of surgery.

As the authors insist (7, 267) that the results of the research are related to the total female sample of the st

The article analyses the correlation between anxiety symptoms and perceived quality of 10 life in women with more than 24 months after undergoing bariatric surgery. As other studies point out, participants had severe and moderate anxiety symptoms (respectively, in this case, 64% and 36%).

Furtherly, the study demonstrates that moderate to severe anxiety symptoms interfere with the perception of quality of life, regardless of weight regain and surgery time.

Evidence of this article are relevant in highlighting the need of an adequate care by multidisciplinary teams, as well as support networks, in order to prevent that anxiety symptoms, together with a decrease of the quality life improvements, can compromise the success of the treatment in the long-term, in particular after 24 months of surgery.

As the authors insist (7, 267) that the results of the research are related to the total female sample of the study participants, some more information on possible gender-specific factors that could influence the results of the research topics would be useful.

Finally, I think “Pezzim” - before [14] - is a typo (2, 64).

udy participants, some more information on possible gender-specific factors that could influence the results of the research topics would be useful.

Finally, I think “Pezzim” - before [14] - is a typo (2, 64).

Author Response

International Journal of Environmental Research and Public Health- Special Issue "Effects of Stress Exposure on Mental Health and Well-Being".

It is with great pleasure that we submit for review the manuscript entitled “Correlation between anxiety symptoms and perception of quality of life in women with more than 24 months after undergoing bariatric surgeryfor consideration by International Journal of Environmental Research and Public Health

We appreciate the reviewer(s) suggestions and clarify that all requested changes have been made. According to the instructions:

Reviewer: 2

The article analyses the correlation between anxiety symptoms and perceived quality of 10 life in women with more than 24 months after undergoing bariatric surgery. As other studies point out, participants had severe and moderate anxiety symptoms (respectively, in this case, 64% and 36%).

Furtherly, the study demonstrates that moderate to severe anxiety symptoms interfere with the perception of quality of life, regardless of weight regain and surgery time.

Evidence of this article are relevant in highlighting the need of an adequate care by multidisciplinary teams, as well as support networks, in order to prevent that anxiety symptoms, together with a decrease of the quality life improvements, can compromise the success of the treatment in the long-term, in particular after 24 months of surgery.

As the authors insist (7, 267) that the results of the research are related to the total female sample of the study participants, some more information on possible gender-specific factors that could influence the results of the research topics would be useful.

R= We appreciate the considerations. A paragraph was inserted at the beginning of the discussion to try to discuss gender-specific factors.

Finally, I think “Pezzim” - before [14] - is a typo (2, 64).

R= We appreciate the considerations and apologize for the typo, correction has been made.

Reviewer 3 Report

This is an interesting topic exploring the anxiety symptoms and QoL after bariatric surgery. 

There is a need to double the check the figures presented in the results, eg. Line 163, the p-value for functional capacity is different from the value presented in Table 2.

Minor grammar and english language amendments are needed to help readers to understand better. 

Author Response

International Journal of Environmental Research and Public Health- Special Issue "Effects of Stress Exposure on Mental Health and Well-Being".

It is with great pleasure that we submit for review the manuscript entitled “Correlation between anxiety symptoms and perception of quality of life in women with more than 24 months after undergoing bariatric surgeryfor consideration by International Journal of Environmental Research and Public Health

We appreciate the reviewer(s) suggestions and clarify that all requested changes have been made. According to the instructions:

Reviewer: 1

  • Abstract: P values should be shown as “p < 0.001” when it is smaller than 0.001. It should not be shown as “p=0.000”.

R= We appreciate the correction and inform you that the changes have been made.

  • Abstract: The statement “between the general physical component of quality of life and anxiety score (β = -0.339; CI -0.899; -0.131; p= 0.010), both independent of weight regain and surgery time.” appears to be misleading because surgery time was not significant (p = 0.294). The statement should be “between the general physical component of quality of life and anxiety score (β = -0.339; CI -0.899; -0.131; p= 0.010), both independent of weight regain.”

R= We thank you for your consideration. In the linear regression test (Table 4), the general physical component of quality of life was considered as the dependent variable and the anxiety score, weight regain and surgery time as independent variables.

Therefore, it was observed that the correlation between anxiety and the general component remained independent of weight regain (p=0.011) and surgery time (p=0.010), even though no correlation was observed between weight regain and general physical component of quality of life, therefore, we did not make this change.

  • Line 29 and 106’: “Body Mass Index” should not be capitalized.

 R= Thanks for the suggestion, changes have been made to the indicated lines.

  • Line 42: “Lee Fool” should be “Le Foll”.

R= Thank you for the correction, the requested adjustment has been made.

  • Line 92: The word “confuse” should be replaced by “affect”.

R= We appreciate the comment, the adjustment has been made.

  • Line 142: “The Statistical Package for Social Science version 24.0 program” should be described as “IBM SPSS Statistics for Windows, Version 24.0 (Armonk, NY, IBM Corp.)”

 R=  We appreciate the correction and inform you that the change has been made.

  • Table 1: In the header row, “DP” should be “SD”.

 R=  Thanks for the correction and we apologize for the error. Change was made.

  • Table 2 and line 162-166: The symbol for the Spearman’s correlation coefficient should be the Greek letter rho (ρ) and not p. In addition, it should not be squared.

 R=  We appreciate the corrections and inform you that the adjustments have been made.

  • Line 172-177, 185-188, and Table 3: P values should be shown as “p < 0.001” when it is smaller than 0.001. It should not be shown as “p=0.000”.

R=We appreciate the corrections and inform you that the changes have been made.

  • Table 3 and 4: The title should be “Linear regression” instead of “Correlation”.

R=Thanks for the suggestions, the changes have been made.

  1. Data analysis: According to the footnote of Table 2, Spearman correlation test rather than Pearson’s correlation test was used. Please correct the inconsistence.

R= Thank you for your consideration, correction has been made.

  • Data analysis: Have the authors evaluated if “current BMI” was a significant factor in the linear regression models in Table 3 and 4?

R= We appreciate the suggestion. The BMI was tested in the model, but there was no change in the correlation. In addition, we emphasize that BMI was not maintained in the linear regression, as this is a cross-sectional study, and BMI is data that is influenced by other variables, such as initial weight. Therefore, it would be more interesting to use BMI if it were a prospective study from the preoperative time. So, for the present cross-sectional study, with the bariatric surgery population, postoperative weight regain seems to have a greater influence on issues such as mental health and quality of life.

  • References: Please check all the references for accuracy and format. For example, reference #8 the page number should be e1916. Reference #26: the page number should be 44. In addition, reference #11: The names of the two authors should be Pilla Della Méa, C., & Peccin, C. See: https://pssaucdb.emnuvens.com.br/pssa/article/view/370

R= Thanks for the corrections, the changes have been made.

Reviewer: 2

The article analyses the correlation between anxiety symptoms and perceived quality of 10 life in women with more than 24 months after undergoing bariatric surgery. As other studies point out, participants had severe and moderate anxiety symptoms (respectively, in this case, 64% and 36%).

Furtherly, the study demonstrates that moderate to severe anxiety symptoms interfere with the perception of quality of life, regardless of weight regain and surgery time.

Evidence of this article are relevant in highlighting the need of an adequate care by multidisciplinary teams, as well as support networks, in order to prevent that anxiety symptoms, together with a decrease of the quality life improvements, can compromise the success of the treatment in the long-term, in particular after 24 months of surgery.

As the authors insist (7, 267) that the results of the research are related to the total female sample of the study participants, some more information on possible gender-specific factors that could influence the results of the research topics would be useful.

R= We appreciate the considerations. A paragraph was inserted at the beginning of the discussion to try to discuss gender-specific factors.

Finally, I think “Pezzim” - before [14] - is a typo (2, 64).

R= We appreciate the considerations and apologize for the typo, correction has been made.

Round 2

Reviewer 1 Report

The authors have adequately responded to some of my comments. However, a number of revisions indicated in the response letter were not incorporated in the manuscript accordingly.

P values in the abstract are still incorrectly presented. P values should be shown as “p < 0.001” when it is smaller than 0.001. It should not be shown as “p=0.000”. In addition, the p values in the abstract seem to be different from those shown in the tables. 

"The domains of quality of life with better perception were in the aspect of 
functional capacity and limitation due to physical aspects (p<0.001). In linear regression, a significant inverse correlation was found between the general emotional component of quality of life and anxiety score (β = -0.546; CI -1.419; -0.559; p=0.000) and ... "

should be 

"The domains of quality of life correlated with better perception were  functional capacity (p=0.003),  pain (p<0.001), general health status (p=0.018), social aspects (p<0.001), and mental health (p<0.001). In linear regression, a significant inverse correlation was found between the general emotional component of quality of life and anxiety score (β = -0.546; CI -1.419; -0.559; p<0.001) and ..."

2. Line 42: “Lee Fool” should be “Le Foll”. 

3. Line 162-165, P values should be shown as “p < 0.001” and not "p < 0.0001". By default, SPSS output only gives three decimal places. 

4. Line 166: "p=<0.0001" should be shown as "p < 0.001".

5. Table 2. P values of < 0.0001 should be shown as < 0.001.

6. The authors should use "." rather than "," for decimal point throughout the manuscript.

Author Response

It is with great pleasure that we submit for review the manuscript entitled “Correlation between anxiety symptoms and perception of quality of life in women with more than 24 months after undergoing bariatric surgeryfor consideration by International Journal of Environmental Research and Public Health

We appreciate the reviewer suggestions and clarify that all requested changes have been made. According to the instructions:

Reviewer: 1

  • The authors have adequately responded to some of my comments. However, a number of revisions indicated in the response letter were not incorporated in the manuscript accordingly.

P values in the abstract are still incorrectly presented. P values should be shown as “p < 0.001” when it is smaller than 0.001. It should not be shown as “p=0.000”. In addition, the p values in the abstract seem to be different from those shown in the tables.

The authors have adequately responded to some of my comments. However, a number of revisions indicated in the response letter were not incorporated in the manuscript accordingly.

P values in the abstract are still incorrectly presented. P values should be shown as “p < 0.001” when it is smaller than 0.001. It should not be shown as “p=0.000”. In addition, the p values in the abstract seem to be different from those shown in the tables.

"The domains of quality of life with better perception were in the aspect of functional capacity and limitation due to physical aspects (p<0.001). In linear regression, a significant inverse correlation was found between the general emotional component of quality of life and anxiety score (β = -0.546; CI -1.419; -0.559; p=0.000) and ... "

should be

"The domains of quality of life correlated with better perception were functional capacity (p=0.003),  pain (p<0.001), general health status (p=0.018), social aspects (p<0.001), and mental health (p<0.001). In linear regression, a significant inverse correlation was found between the general emotional component of quality of life and anxiety score (β = -0.546; CI -1.419; -0.559; p<0.001) and ..."

R= We appreciate the correction and inform you that the changes have been made. The change in the text, according to the values presented in table 2, was as follows:

“The domains of quality of life correlated with better perception were pain (p<0.001), functional capacity (p=0.013), general health status (p=0.018), social aspects (p<0.001), and mental health (p<0.001). In linear regression, a significant inverse correlation was found between the general emotional component of quality of life and anxiety score (β = -0.546; CI -1.419; -0.559; p<0.001) and…”

  • Line 42: “Lee Fool” should be “Le Foll”.

R= Thank you for the correction, the requested adjustment has been made.

  1. Line 162-165, P values should be shown as “p < 0.001” and not "p < 0.0001". By default, SPSS output only gives three decimal places.

R= We appreciate the comment, the adjustment has been made.

  1. Line 166: "p=<0.0001" should be shown as "p < 0.001".

 R=  We appreciate the correction and inform you that the change has been made.

  1. Table 2. P values of < 0.0001 should be shown as < 0.001.

 R=  We appreciate the corrections and inform you that the adjustments have been made.

  1. The authors should use "." rather than "," for decimal point throughout the manuscript.

 R=  Thanks for the correction and we apologize for the error. Change was made.
